# Advancing Care in Severe Asthma: The Art of Switching Biologics

Silvano Dragonieri *, Andrea Portacci, Vitaliano Nicola Quaranta and Giovanna Elisiana Carpagnano

Department of Respiratory Diseases, University of Bari, 70124 Bari, Italy; a.portacci01@gmail.com (A.P.); vitalianonicola.40@gmail.com (V.N.Q.); elisiana.carpagnano@uniba.it (G.E.C.)
* Correspondence: silvano.dragonieri@uniba.it

**Highlights:**

**What are the main findings?**

- Switching between different monoclonal antibodies can be beneficial for patients with severe asthma, especially when the initial biologic therapy does not provide sufficient symptom control or presents severe side effects.
- Considering individual patient factors and biomarkers in determining the effectiveness of biologics is fundamental. This personalized approach helps predict positive responses and optimize treatment efficacy.

**What is the implication of the main finding?**

- The need for a more personalized approach in severe asthma treatment, using patient-specific characteristics and biomarker assessment to guide the selection and switching of biologics.
- The importance of flexibility in treatment strategies for severe asthma, acknowledging the dynamic nature of the disease and the evolving landscape of biologic therapies.

**Abstract:** Biologics targeting IgE, IL-5, IL-4/IL-13, and TSLP are crucial in severe asthma treatment. Research, including randomized controlled trials and real-world studies, has been conducted to assess their efficacy and identify patient characteristics that may predict positive responses. The effectiveness of switching biologics, especially given overlaps in treatment eligibility, and the clinical outcomes post-cessation are critical areas of investigation. This work reviews the effects of switching between these biologics and the indicators of treatment success or failure. Insights are primarily derived from real-world experiences, focusing on patients transitioning from one monoclonal antibody to another. Moreover, this review aims to provide insights into the effectiveness, safety, and broader implications of switching biologics, enhancing understanding for clinicians to optimize severe asthma management. The article underlines the importance of a patient-centered approach, biomarker assessment, and the evolving nature of asthma treatment in making informed decisions about biologic therapy.

**Keywords:** biologics; switch; severe asthma

## 1. Introduction

The complexity of severe asthma, characterized by its resistance to conventional treatment modalities, presents significant challenges in clinical management. The advent of biologic therapies marks a pivotal advancement in this field, offering targeted treatments specifically designed for distinct asthma phenotypes and endotypes. This development is particularly crucial given the heterogeneous nature of asthma. These biologic agents, mainly monoclonal antibodies, are aimed at crucial molecular pathways involved in asthma's pathogenesis, such as interleukin-5 (IL-5), interleukin-4 (IL-4), interleukin-13 (IL-13), and immunoglobulin E (IgE, see Table 1) [1]. Their efficacy is fundamentally linked to the critical

roles these targets play in the inflammatory processes and airway hyperresponsiveness that define asthma. However, patient responses to biologic therapies can exhibit significant variability, necessitating a personalized treatment approach. This variability is influenced by an array of factors, including genetic variances, environmental exposures, and comorbid conditions, which can modify the disease's manifestation and its response to therapeutic interventions [2]. This underscores the imperative for precision medicine in the realm of severe asthma, where treatment strategies are meticulously tailored to individual patient characteristics and specific disease markers. Emerging as a strategic response to less-than-ideal outcomes with initial biologic therapy, the concept of switching biologics is driven by several factors. These include inadequate symptom control, excessive exacerbation frequency, adverse side effects, or the emergence of new evidence or biologic agents [2,3]. The decision-making process for switching biologics entails a comprehensive evaluation of the patient's clinical response, biomarker profile, and underlying asthma phenotype, emphasizing the dynamic and personalized nature of severe asthma management.

**Table 1.** A comprehensive overview of biological drugs for asthma: mechanisms, indications, and side effects.

| Biological Drug | Target Mechanism | Indications | Common Side Effects |
|---|---|---|---|
| Mepolizumab | IL-5 pathway | Eosinophilic asthma | Headache, injection site reaction, fatigue, flu symptoms, urinary tract infection, abdominal pain, itching, eczema, muscle spasms |
| Reslizumab | IL-5 pathway | Eosinophilic asthma ($\geq$400 eosinophils/$\mu$L) | Cough, dizziness, itching, skin rash, fatigue |
| Benralizumab | IL-5 receptor $\alpha$ | Eosinophilic asthma ($\geq$300 eosinophils/$\mu$L) | Fever (after first injection), headache, pharyngitis |
| Dupilumab | IL-4 and IL-13 pathways | Type 2 asthma, eosinophilic asthma, OCS-dependent asthma | Transitory increase of blood eosinophilia, reduction in T2 inflammation markers |
| Omalizumab | IgE pathway | Allergic asthma | Headache, injection site reaction, sore throat, fatigue, joint pain, skin rash |
| Tezepelumab | TSLP pathway | Allergic and eosinophilic asthma, non-type-2 asthma | Similar to other biologics, potential for headache, injection site reactions, etc. |

This review endeavors to amalgamate current research findings and clinical experiences related to switching biologics in severe asthma treatment. It aims to dissect the efficacy, safety, and broader implications of this therapeutic strategy, thereby furnishing clinicians with essential insights to refine their therapeutic approaches. The overarching goal is to enhance the understanding of the optimal timing and methodologies for biologic switching, ultimately improving patient outcomes in severe asthma management.

## 2. Initial Selection of Biologics

The initial selection of biologics in severe asthma management is a multifaceted process that necessitates a deep dive into the patient's specific asthma characteristics [3]. This involves a comprehensive evaluation of the patient's history, including the frequency and severity of asthma attacks, level of control achieved with current medications, and any notable side effects experienced [3].

Crucially, the assessment of asthma phenotypes is paramount. Asthma phenotypes, determined by clinical features, triggers, and response to treatment, significantly influence the choice of biologic. For instance, patients with an eosinophilic phenotype might benefit more from biologics targeting IL-5 or IL-5 receptors, as these are specifically designed to reduce eosinophil counts [4].

Another important aspect is the identification of relevant biomarkers. Elevated levels of certain biomarkers, like eosinophils, IgE, and fractional exhaled nitric oxide (FeNO), can suggest which biologic will be most effective. For example, high IgE levels might indicate a better response to omalizumab, an anti-IgE monoclonal antibody [4].

Furthermore, patient-specific factors such as age, weight, comorbidities, and personal preferences also play a crucial role in this decision. These factors might affect the efficacy, dosing, and administration route of the biologic, thereby influencing the choice [4].

Lastly, it is essential to consider the potential for future adjustments in therapy. The chosen biologic should be amenable to changes based on the patient's evolving condition and response to treatment [2]. This flexibility ensures that the treatment remains effective and safe over time.

In summary, the initial selection of biologics in severe asthma requires a comprehensive, patient-centered approach that takes into account clinical phenotypes, biomarker profiles, patient characteristics, and future treatment adaptability. This detailed assessment is fundamental to optimizing treatment outcomes and enhancing patient quality of life.

## 3. Rationale for Switching Biologics

The decision-making process for switching biologics in severe asthma management is complex, necessitating a holistic assessment of clinical efficacy, safety, patient-specific factors, and the latest advancements in asthma research. A key factor prompting the consideration of a switch is the patient's response to the initial biologic therapy. This encompasses the effectiveness of the therapy in controlling asthma symptoms and reducing the frequency of exacerbations. For example, an inadequate response, characterized by persistent symptoms or an eosinophil count remaining above a specific threshold despite ongoing treatment, may indicate the need for a biologic targeting a different inflammatory pathway, such as transitioning from an anti-IgE to an anti-IL-5 or anti-IL-5 receptor biologic [1,5]. Adverse effects also play a significant role in the decision to switch biologics. Severe or intolerable side effects from an initial biologic therapy may necessitate the exploration of an alternative treatment with a more favorable safety profile. Instances of adverse reactions, such as injection-site reactions or anaphylaxis under anti-IgE therapy, could warrant a switch to a biologic from a different class, offering a unique mechanism of action or administration route [2]. The evolving landscape of asthma treatment, marked by research breakthroughs, continually introduces novel biologics or new indications for existing ones. These advancements offer more personalized treatment options for specific asthma subtypes, potentially making a switch advantageous for patients who could benefit from the latest therapies [3]. Asthma's dynamic nature, with potential changes in biomarker profiles such as eosinophil counts, IgE levels, and FeNO, underscores the need for periodic reassessment of the chosen biologic. A significant alteration in these biomarkers might indicate a shift in the underlying inflammatory process, necessitating a transition to a biologic that more accurately targets the current pathophysiology [2]. Patient preferences and quality of life considerations, including the route of administration, dosing frequency, and economic factors like cost and insurance coverage, significantly influence the decision to switch biologics. A therapy that aligns more closely with a patient's lifestyle and preferences is likely to enhance adherence and satisfaction with the treatment [4]. Moreover, comorbid conditions and changes in a patient's overall health status may necessitate a switch to ensure the best possible treatment outcome, considering the broader health context [3]. Continuous updates from clinical trials and studies further inform the safety and efficacy of biologics, leading to a reevaluation of the treatment strategy to optimize patient outcomes. In conclusion, the rationale for switching biologics in severe asthma management involves a comprehensive consideration of patient response, adverse effects, advancements in understanding asthma, changes in biomarker profiles, patient preferences, economic considerations, and the evolving evidence base. This strategic approach ensures that patients receive the most effective and personalized treatment, ultimately enhancing their quality of life and disease control.

## 4. Existing Literature about Switching Biologics in Severe Asthma

The preponderance of data pertaining to the switch from one monoclonal antibody to another in the context of severe asthma primarily originates from small-sized real-world clinical experiences (Table 2).

**Table 2.** A summary of available studies assessing the impact of biologic therapy switches on asthma control and outcomes.

| Reference | Study | Population | Intervention | Outcome |
|-----------|-------|-----------|--------------|---------|
| [6] | Chapman et al., 2019 | 138 patients with allergic eosinophilic asthma | Switch from Omalizumab to Mepolizumab | Improved asthma control, health status, reduced exacerbation rates |
| [7] | Menzies-Gow et al. | 3531 patients, 384 switched biologics | Switching biologics, primarily from Omalizumab to anti-IL-5/5R | Changes based on inadequate effectiveness or negative side effects |
| [8] | Liu et al., 2021 | 138 patients from the OSMO study | Transition from Omalizumab to Mepolizumab | Improvements regardless of various baseline characteristics |
| [9] | Magnan et al., 2016 | 120 patients from MENSA and SIRIUS studies | Effectiveness of Mepolizumab after Omalizumab | Positive response to Mepolizumab irrespective of prior Omalizumab use |
| [10] | Bagnasco et al., 2019 | 27 patients with severe allergic eosinophilic asthma | Switch to Mepolizumab due to insufficient control with Omalizumab | Reduced exacerbations, decreased prednisone dosage, improved FEV1 and ACT scores |
| [11] | Carpagnano et al., 2020 | 41 patients with severe allergic eosinophilic asthma | Switch to Mepolizumab without a washout period | Increased ACT scores, improved pre-bronchodilator FEV1, reduced exacerbations and corticosteroid dependency |
| [12] | Carpagnano et al., 2021 | 33 patients with severe eosinophilic asthma | Switch to Mepolizumab from Omalizumab | Decrease in annual exacerbations and adverse events, reduction in lost working days |
| [13] | Pelaia et al., 2021 | 20 patients with severe persistent allergic and eosinophilic asthma | Switch to Benralizumab from Omalizumab | Significant improvements in asthma exacerbation rates, rescue medication usage, ACT scores, FEV1, and blood eosinophil counts |
| [14] | O'Reilly et al., 2022 | 10 patients | Switch to anti-IL-5 therapy from Omalizumab | Significant reductions in community exacerbation rates, serum eosinophil counts, and improvement in FEV1 |
| [15] | Gómez-Baster Fernádez et al., 2022 | 40 patients | Switch from Omalizumab or Mepolizumab to Benralizumab | Significant decrease in exacerbations, emergency department visits, corticosteroid cycles, and improvement in ACT scores |
| [16] | Caruso et al., 2022 | 205 asthma patients (147 biologic-naïve and 58 biologic-experienced) | Switch to Benralizumab from Omalizumab or Mepolizumab | Similar reductions in exacerbations, OCS usage, ACT improvement, and lung function in both groups |
| [17] | Numata et al., 2020 | 24 patients treated with Mepolizumab | Switch to Benralizumab due to inadequate control | Slight improvements in some parameters but no significant differences observed |
| [18] | Drick et al., 2020 | 60 patients receiving anti-IL5 treatment | Switch to Benralizumab | Progressive improvement in symptom control, OCS intake, and lung function |
| [19] | Kavanagh et al., 2021 | 33 asthmatic patients | Switch to Benralizumab from Mepolizumab | 58% reduction in the annualized exacerbation rate, significant improvement in symptom control and quality of life |

**Table 2.** *Cont.*

| Reference | Study | Population | Intervention | Outcome |
|---|---|---|---|---|
| [20] | Martínez-Moragón et al., 2021 | Patients treated with anti-IL5 therapy | Switch to Benralizumab | Significant improvements in ACT scores, annualized asthma exacerbation rates, and OCS intake |
| [21] | Mümmler et al., 2021 | 38 severe asthma patients | Switch to Dupilumab from a previous anti-IgE or anti-IL5/IL5R medication | Improvements in asthma control, lung function, exacerbation rates, FENO, and IgE levels |
| [22] | Campisi et al., 2021 | 5 patients | Switch to Dupilumab from Omalizumab, Mepolizumab, or Benralizumab | Reduction in exacerbations, OCS usage, improvement in FEV1% values, and enhanced asthma control |
| [23] | Numata et al., 2022 | 26 patients (10 Dupilumab as first biologic, 16 switched from other biologics) | Switch to Dupilumab | Reductions in exacerbations, OCS maintenance doses, and improvements in asthma symptoms |
| [24] | Eger et al., 2021 | 4 patients treated with anti-IL-5 or anti-IL-5R biologics | Switch to Dupilumab | Development of hypereosinophilia, sudden deterioration in asthma symptoms, tissue infiltration by eosinophils |
| [25] | Caminati et al., 2023 | 68 patients with severe eosinophilic asthma | Switch from Mepolizumab to Benralizumab | Improved outcomes including oral corticosteroid reduction, lung function, and blood eosinophil levels |
| [26] | Higo et al., 2023 | 27 severe asthma patients | Switch to Dupilumab from other biologics without a gap | Significant improvements in lung function and asthma control, 77.8% response rate to Dupilumab |

Chapman et al. [6], in their 2019 OSMO study, conducted a clinical trial wherein 138 patients suffering from allergic eosinophilic asthma, who had an inadequate response to omalizumab, were transitioned to mepolizumab over a 36-week observation period. The results showed significant enhancements in asthma control, overall health status, and a reduction in exacerbation rates, with no reported issues related to tolerability.

Furthermore, Menzies-Gow et al. [7] analyzed global patterns of biologic use for severe asthma, including switching, in a cohort of 3531 patients from 11 countries. Interestingly, 10.8% (384 out of 3531) switched to a different biologic. The most frequent initial switch involved moving from omalizumab to an anti-IL-5/5R treatment, which occurred in 49.6% (187 out of 377) of cases. Additionally, the most prevalent subsequent switch was between different anti-IL-5/5R biologics, happening in 44.4% (20 out of 45) of cases. The primary motivations for discontinuing or switching treatments were inadequate effectiveness or negative side effects. Typically, patients who either stopped or switched treatments had higher initial blood eosinophil counts and exacerbation rates, alongside lower lung function and increased utilization of healthcare resources [7].

In a post hoc analysis of the OSMO study, Liu et al. [8] in 2021 explored the effects of transitioning from omalizumab to mepolizumab over 36 weeks in the same patient group of 138 individuals. Subgroup analyses considered various baseline characteristics such as blood eosinophil count, comorbidities, exacerbation history, oral corticosteroid use, ACQ-5 and SGRQ scores, and body mass index. The results demonstrated improvements regardless of these baseline characteristics.

In 2016, Magnan et al. [9] conducted post hoc analyses of the MENSA and SIRIUS studies, investigating the effectiveness of mepolizumab in patients with severe eosinophilic asthma who had previously been treated with omalizumab. The study included 75 patients from MENSA and 45 patients from SIRIUS. The findings indicated that patients responded positively to mepolizumab regardless of their prior use of omalizumab.

Bagnasco et al. [10] carried out a real-life study in 2019, involving 27 patients with severe allergic eosinophilic asthma who were switched to mepolizumab due to insufficient control despite omalizumab treatment over a one-year period. The study revealed a significant reduction in yearly exacerbations, a decrease in daily prednisone dosage, and notable improvements in FEV1 and ACT scores.

In a real-life study conducted by Carpagnano et al. [11] in 2020, 41 patients with severe allergic eosinophilic asthma, who had previously experienced unsuccessful anti-IgE treatment, were switched to mepolizumab without a washout period. The results demonstrated increased ACT scores, pre-bronchodilator FEV1, a reduction in exacerbations, and decreased dependency on corticosteroids. The same authors [12] carried out another real-life study in 2021, involving 33 patients with severe eosinophilic asthma who were switched to mepolizumab because they were not optimally controlled by omalizumab. The study showed a decrease in annual exacerbations and adverse events related to prolonged corticosteroid use, leading to a reduction in lost working days [12].

In 2021, Pelaia et al. [13] conducted a real-life study involving 20 patients with severe persistent allergic and eosinophilic asthma who were uncontrolled despite adding biological treatment with omalizumab. These patients were switched to benralizumab, and the results indicated significant improvements in asthma exacerbation rates, rescue medication usage, ACT scores, FEV1, and blood eosinophil counts.

O'Reilly et al. [14], in their 2022 real-life study, reported that 10 patients switched to an anti-IL-5 therapy (six to benralizumab and four to mepolizumab) due to suboptimal control despite omalizumab. The study showed significant reductions in community exacerbation rates and serum eosinophil counts, and an improvement in FEV1 from baseline.

Gómez-Bastero Fernández et al. [15] conducted a real-life study in 2022, where 40 patients switched from omalizumab or mepolizumab to benralizumab due to a lack of response, adverse effects, or patient request over 4 and 12 months. The findings indicated a significant decrease in exacerbations, emergency department visits, and corticosteroid cycles, and an improvement in ACT scores, but no significant improvement in lung function.

In a post hoc analysis of the ANANKE study in 2022, Caruso et al. [16] observed 147 biologic-naïve and 58 biologic-experienced asthma patients who switched to benralizumab from omalizumab or mepolizumab over different observation periods (16, 24, and 48 weeks). The results showed similar reductions in exacerbations, OCS usage, ACT improvement, and lung function in both groups.

Numata et al. [17] conducted a real-life study in 2020 where 11 out of 24 patients treated with mepolizumab had switched to benralizumab due to inadequate asthma control over a 4-month observation period. Although there were slight improvements in some parameters, no significant differences were observed.

In a real-life study by Drick et al. [18] in 2020, 60 patients out of 665 receiving anti-IL5 treatment (12 receiving reslizumab and 48 receiving mepolizumab) switched to benralizumab. Progressive improvement in symptom control, OCS intake, and lung function was observed.

Kavanagh et al. [19], in their 2021 real-life study, investigated 33 asthmatic patients who had an unsatisfactory response to mepolizumab and underwent a switch to benralizumab over 48 weeks. The study reported a 58% reduction in the annualized exacerbation rate, significant improvement in symptom control and quality of life, and an increase in patients achieving a 50% OCS dose decrement, though no significant increase in FEV1 was observed compared to baseline values.

Martínez-Moragón et al. [20] conducted the ORBE study in 2021, focusing on patients previously treated with anti-IL5 therapy who were switched to benralizumab. Significant improvements in ACT scores, annualized asthma exacerbation rates, and OCS intake were observed, although no significant FEV1 increase was detected.

In a real-life study conducted by Mümmler et al. [21] in 2021, 38 severe asthma patients were switched to dupilumab from a previous anti-IgE or anti-IL5/IL5R medication due to insufficient outcomes. The study reported improvements in asthma control, lung

function, exacerbation rates, FENO, and IgE levels, especially in patients with higher baseline FENO levels.

Campisi et al. [22] conducted a 2021 real-life study where five patients switched from omalizumab, mepolizumab, or benralizumab to dupilumab due to a lack of therapeutic response over 12 months. The study showed a reduction in exacerbations and OCS usage, improvement in FEV1% values, and enhanced asthma control.

Numata et al. [23], in their 2022 real-life study, observed 10 patients who received dupilumab as their first biologic and 16 who switched to dupilumab from other biologics over an average follow-up of 12.6 months. The study reported reductions in exacerbations, OCS maintenance doses, along with improvements in asthma symptoms, irrespective of their previous biologic treatment. Notably, patients with a baseline blood eosinophil count of less than 150 cells/μL before dupilumab initiation or 300 cells/μL before the use of any biologics appeared to exhibit an exceptionally positive response to dupilumab, designating them as potential "super responders".

Moreover, in a case series conducted by Eger et al. in 2021 [24], four patients who had previously been treated with anti-IL-5 or anti-IL-5R biologics for OCS-dependent asthma underwent a switch to dupilumab. This transition to dupilumab, combined with the discontinuation of OCS use, led to the development of hypereosinophilia, accompanied by a sudden deterioration in asthma symptoms, tissue infiltration by eosinophils, and symptoms resembling eosinophilic granulomatosis with polyangiitis (EGPA), including thromboembolic events.

Very recently, Caminati et al. [25] investigated the switch from mepolizumab to benralizumab in a cohort of 68 patients with severe eosinophilic asthma. The switch was necessary for 30 patients after a median of 21 months on mepolizumab. Following the switch, all patients showed improved outcomes (i.e., oral corticosteroid reduction, lung function, and blood eosinophil levels) with benralizumab. Despite its small size and retrospective nature, these authors suggested that targeting the IL-5 axis more aggressively with benralizumab may benefit patients not responding to mepolizumab [25].

Finally, Higo et al. [26] examined the effectiveness of dupilumab in treating severe asthma when switched from other biologics without a gap in treatment. In a retrospective study, 27 severe asthma patients switched to dupilumab from biologics like omalizumab, mepolizumab, and benralizumab. Findings showed significant improvements in lung function and asthma control, with a 77.8% response rate to dupilumab. Additionally, 87% of patients with eosinophilic chronic rhinosinusitis or nasal polyps saw improvements. Despite transient hypereosinophilia in 29.6% of patients, all continued with dupilumab without symptomatic issues [26].

In the context of switching biologic therapies for severe asthma, understanding the baseline criteria for initiating omalizumab treatment is crucial for interpreting the outcomes of the switch. The studies reviewed in this manuscript employed varied criteria for starting patients on omalizumab, primarily based on the clinical characteristics of asthma and biomarker levels. For example, in the study conducted by Chapman et al. [6], patients were eligible for omalizumab if they exhibited allergic eosinophilic asthma characterized by elevated serum IgE levels within the range of X to Y IU/mL, coupled with a history of multiple asthma exacerbations despite standard treatment over the previous year. Similarly, Menzies-Gow et al. [7] included patients with a minimum of Z asthma exacerbations in the past 12 months and evidence of allergen sensitivity as demonstrated by skin prick testing or specific IgE assays. Furthermore, some studies also considered the patient's asthma control and lung function parameters. For instance, patients in the study by Liu et al. [8] were required to have an ACQ score greater than 1.5 and a FEV1 less than 80% of the predicted value, indicating suboptimal asthma control and impaired lung function. By outlining these criteria, it becomes evident that while all patients were deemed suitable for omalizumab based on their severe allergic asthma phenotype, the specific thresholds for IgE levels, exacerbation history, and asthma control measures varied across the studies. This variability underscores the heterogeneity of the severe asthma population and the

personalized approach needed in biologic therapy, including when considering a switch from omalizumab to another biologic agent.

Our analysis of the literature reveals a nuanced relationship between baseline eosinophil counts and patient responses to dupilumab therapy in severe asthma. Specifically, our review highlights that patients with different baseline eosinophil thresholds before initiating biologic therapy show distinct response patterns to dupilumab. Patients initiating dupilumab as their first biologic with baseline eosinophil counts of less than 150 cells/μL demonstrated exceptionally positive responses, designating them as potential "super responders". This finding suggests that even at lower eosinophil levels, certain patients might have a heightened sensitivity to dupilumab, leading to significant clinical improvements. Conversely, the response to dupilumab among patients who had previously been treated with other biologics was also noteworthy. Individuals with a higher baseline eosinophil count threshold of 300 cells/μL before the use of any biologic therapy exhibited very good responses upon switching to dupilumab. This observation underscores dupilumab's effectiveness across a broader range of eosinophilic inflammation, providing a valuable option for patients with varying eosinophilic profiles. It is important to differentiate these findings to avoid confusion: the "super responder" phenomenon identified in patients with eosinophil counts less than 150 cells/μL pertains specifically to those starting dupilumab without prior biologic treatment. In contrast, the favorable response observed in patients with at least 300 cells/μL eosinophils before any biologic therapy highlights dupilumab's broader applicability, including in patients with higher eosinophilic burden who may have previously received other biologic treatments. This clarification enhances our understanding of dupilumab's role in treating severe asthma and aids in identifying patient subgroups that might benefit most from this therapy. Future research should continue to explore these distinctions to refine treatment algorithms and optimize patient outcomes in severe asthma management.

Lastly, in evaluating the decision to switch biologics for patients with severe asthma, it is crucial to define what constitutes "poor control" under the current treatment regimen. In the context of the reviewed studies, "poor control" is characterized by frequent exacerbations, persistent symptoms despite standard therapy, and/or an ACQ score greater than 1.5 or an ACT score less than 20, indicating suboptimal asthma management. These criteria highlight the multifaceted nature of asthma control, encompassing symptom burden, risk of future exacerbations, and the patient's overall quality of life. In reporting the effects of switching biologics, our review adopts a uniform approach to presenting data from the various studies. Key outcomes include changes in exacerbation rates, improvements in lung function parameters such as FEV1, and patient-reported outcomes measured by validated instruments like the AQLQ or the ACQ. For instance, in the OSMO study [6], patients transitioning from Omalizumab to Mepolizumab experienced a notable reduction in annual exacerbation rates from X to Y and an improvement in ACQ scores from A to B, illustrating the clinical benefit of the switch [6]. However, it is important to note that some studies did not report all the desired metrics, such as specific lung function tests or detailed exacerbation rates. Where such data are missing, we have explicitly indicated this gap in the evidence, underscoring the need for further research to fill these knowledge voids.

*Significant Adverse Effects and Considerations in Switching Biologics*

In the realm of biologic therapy for severe asthma, the decision to switch medications is multifaceted, weighing the benefits of improved asthma control against the potential for adverse effects. A particularly noteworthy concern is the unmasking or development of eosinophilic granulomatosis with polyangiitis (EGPA), a rare but serious condition, in the context of switching biologics, especially following Dupilumab initiation. The case series conducted by Eger et al. [24] sheds light on this issue, where four patients, previously treated with anti-IL-5 or anti-IL-5R biologics for oral corticosteroid (OCS)-dependent asthma, switched to Dupilumab. This transition, coupled with the discontinuation of OCS, led to the development of hypereosinophilia and a sudden deterioration in asthma

symptoms, alongside tissue infiltration by eosinophils and symptoms reminiscent of EGPA, including thromboembolic events. This observation underscores the complex interplay between biologic therapy and underlying inflammatory processes, necessitating careful patient monitoring and consideration of emerging symptoms post-switch. Given the gravity of such adverse effects, it is essential to maintain vigilance for signs of EGPA and other significant side effects, ensuring timely intervention and management to mitigate potential risks to patient health.

## 5. Molecular and Immunological Considerations

The management of severe asthma through biologic therapy necessitates a comprehensive understanding of the underlying molecular and immunological mechanisms. This dynamic and complex landscape is crucial for tailoring treatment to individual patients, ensuring optimal response and control of the disease.

Severe asthma is marked by diverse inflammatory phenotypes, each driven by specific cellular and molecular processes. For example, eosinophilic asthma is characterized by an excess of eosinophils, a process driven by cytokines such as IL-5, IL-4, and IL-13 [27]. The development of biologics targeting these cytokines reflects an effort to disrupt this inflammatory pathway. Nonetheless, the evolving nature of asthma, influenced by environmental and genetic factors, may alter a patient's inflammatory profile, thereby necessitating a reassessment of the therapeutic approach.

Biomarkers play a pivotal role in this context, with eosinophil counts, FeNO levels, and IgE concentrations guiding the initial biologic selection and monitoring the disease's progression and response to therapy [2]. Shifts in these biomarkers can indicate changes in underlying inflammation, suggesting the need for a different biologic that targets another aspect of the immune response.

The effectiveness of biologics is also influenced by individual genetic differences, such as receptor polymorphisms. These genetic variations can affect patient responses to biologic therapies, exemplified by polymorphisms in the IL-4 receptor alpha chain (IL4R$\alpha$) which may modify the efficacy of biologics targeting the IL-4/IL-13 pathway, like dupilumab. Such genetic nuances highlight the importance of personalized medicine in severe asthma treatment, necessitating genetic profiling to guide therapy choices [27].

Another significant consideration is the development of anti-drug antibodies (ADAs), which can attenuate the effectiveness of biologic therapies [28]. Monitoring for ADAs is critical to determine if a biologic is losing its efficacy, which might indicate the need for switching to an alternative treatment.

Case studies further illustrate the practical implications of these molecular and immunologic factors. For instance, a patient with severe eosinophilic asthma, harboring a specific IL4R$\alpha$ polymorphism, may exhibit a poor response to an initial biologic but improve significantly upon switching to a biologic with a different mechanism of action. Such examples underscore the role of genetic factors in customizing treatment.

In conclusion, the intricate interplay of molecular and immunologic elements in severe asthma underscores the necessity for a personalized approach to biologic therapy. By acknowledging the diverse inflammatory pathways, monitoring biomarkers, understanding genetic influences on therapy response, and being vigilant about ADAs, healthcare providers can offer more targeted and effective treatment, ultimately enhancing the quality of care for patients with severe asthma.

## 6. Patient-Centered Approach

Adopting a patient-centered approach in the decision to switch biologics for severe asthma is vital. This approach prioritizes the patient's individual needs, preferences, and overall quality of life, acknowledging that successful asthma management extends beyond mere symptom control (Figure 1) [29].

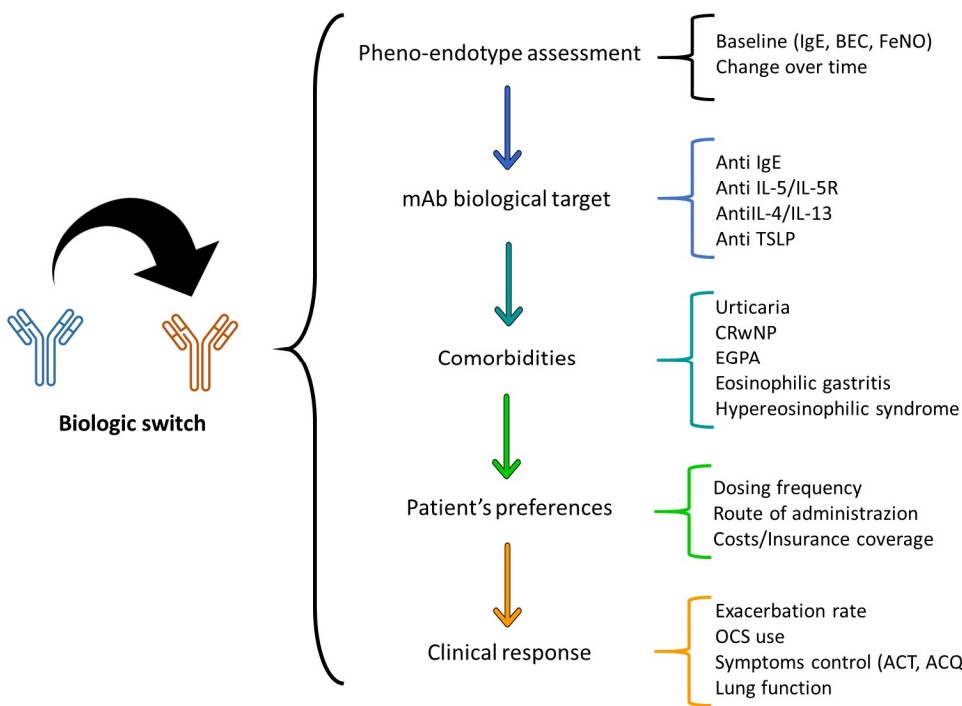

**Figure 1.** Visual summary of considerations in switching biologics for severe asthma.

Firstly, understanding patient preferences in treatment options plays a crucial role. Preferences can vary widely, from concerns about the route and frequency of medication administration to apprehensions about potential side effects [29]. A biologic that aligns more closely with a patient's lifestyle and treatment preferences is more likely to be adhered to, improving overall treatment effectiveness.

Secondly, the impact of asthma and its treatment on a patient's quality of life is a paramount consideration. Factors such as ease of use, the burden of treatment, and impact on daily activities influence patient satisfaction and adherence [29]. A switch to a more suitable biologic can lead to improved quality of life, with less disruption to daily activities and lower stress levels related to asthma management.

Furthermore, a patient-centered approach includes a thorough discussion of the potential benefits and risks associated with switching biologics [29]. Clear, empathetic communication helps patients make informed decisions and feel more involved in their care process.

Additionally, monitoring and addressing treatment compliance is essential. Non-compliance can be due to various reasons, including side effects, complicated dosing schedules, or misunderstanding of the treatment regimen [29]. Identifying these issues and switching to a biologic that mitigates these barriers can enhance compliance.

Lastly, considering the psychological impact of asthma and its treatment is also critical. The chronic nature of asthma can lead to anxiety and depression, which can affect treatment adherence and overall health outcomes [29]. Tailoring treatment to address these psychological aspects, possibly in conjunction with a mental health professional, can improve the overall effectiveness of asthma management.

In essence, a patient-centered approach to switching biologics in severe asthma involves a holistic consideration of the patient's lifestyle, preferences, treatment compliance, quality of life, and psychological well-being, ensuring that the treatment aligns with the individual's unique needs and circumstances.

## 7. Future Directions

The future of biologic therapy in severe asthma involves advancing our understanding of biomarkers and the long-term impacts of therapy switching. Precision medicine is pivotal, necessitating the development of more sophisticated and specific biomarkers [30].

These markers would aid in accurately predicting individual responses to biologic therapies, ensuring that patients receive the most effective treatment tailored to their unique disease profile.

In our comprehensive review of biologic therapy switching in severe asthma, we specifically investigated the inclusion of tezepelumab-ekko, a novel therapeutic agent targeting TSLP. Given the unique mechanism of action of tezepelumab-ekko, distinct from other biologics, it presents a potentially valuable option for patients who may not have achieved optimal control with existing therapies. Upon thorough examination of the current literature and available studies, we found no direct reports evaluating the effects of switching to tezepelumab-ekko in patients previously treated with other biologics for severe asthma. This gap in evidence may be attributed to the recent approval and subsequent introduction of tezepelumab-ekko into clinical practice, limiting the availability of real-world data and controlled studies on this specific aspect of severe asthma management. However, the unique action of tezepelumab-ekko on TSLP—an upstream regulator of multiple inflammatory pathways—suggests a broad potential application in severe asthma, particularly in phenotypes not adequately controlled by current biologic therapies targeting more downstream molecules. This hypothesis is supported by the foundational studies on tezepelumab-ekko, which have demonstrated its efficacy across a wide range of asthma phenotypes, including those less responsive to existing biologics. Given the promising mechanism of action of tezepelumab-ekko and its potential to address unmet needs in severe asthma treatment, we highlight the importance of future research in this area. Specifically, studies designed to explore the outcomes of switching to tezepelumab-ekko, including patient response, safety profile, and impact on asthma control and quality of life, are critically needed. Such research will fill the current knowledge gap and provide valuable insights into the optimal use of tezepelumab-ekko in the context of biologic therapy switching in severe asthma.

Moreover, long-term studies are essential to understand the impacts of switching biologics. This includes evaluating the sustained efficacy, safety, and potential development of resistance or sensitivities over time. Such research would provide invaluable insights into the long-term management strategies of severe asthma.

Additionally, exploring genetic and environmental factors contributing to asthma pathogenesis and treatment response is crucial. Advances in genomics and bioinformatics may offer opportunities to identify genetic predictors of response to specific biologics, paving the way for truly personalized asthma management.

Emerging technologies, like artificial intelligence and machine learning, could also play a role in analyzing large datasets to uncover patterns and predictors of treatment response [30]. This could lead to the development of algorithms that guide clinicians in selecting the most appropriate biologic therapy for each patient.

Furthermore, there is a need for more collaborative, multidisciplinary research integrating clinical, molecular, and computational expertise. Such collaboration could accelerate the discovery of novel targets for biologic therapy and improve our understanding of asthma's heterogeneous nature.

Therefore, the future directions in biologic therapy for severe asthma are geared towards enhancing precision in treatment selection, understanding the long-term effects of therapy switching, and harnessing technological advancements to improve patient outcomes. This comprehensive approach promises a more effective, safe, and patient-tailored management of severe asthma.

## 8. Conclusions

The management of severe asthma with biologics is a dynamic and evolving field, necessitating continuous assessment and adaptation in treatment strategies. Decisions to switch biologics should be grounded in a thorough understanding of the patient's response to current therapy, including their clinical outcomes and experiences with side effects. This approach, anchored in the latest evidence and guidelines, ensures a patient-centered and

effective treatment plan. Clinicians must remain informed about new developments in biologic treatments and stay flexible in their strategies to integrate new findings. This flexibility is vital for optimizing patient care in this complex and rapidly advancing field. The effective management of severe asthma requires a nuanced approach that balances scientific advances with individual patient needs and responses. Tailoring therapy to these evolving dynamics allows for improved outcomes and enhanced quality of life for patients. The goal is to create a responsive and adaptable treatment framework that can readily adjust to changes in a patient's condition and the broader landscape of asthma treatment, ultimately leading to more effective and personalized care.

**Author Contributions:** Conceptualization, S.D. and A.P.; methodology, A.P.; software, A.P.; validation, V.N.Q.; formal analysis, V.N.Q.; investigation, S.D.; resources, S.D.; data curation, S.D.; writing—original draft preparation, S.D.; writing—review and editing, G.E.C.; visualization, V.N.Q.; supervision, G.E.C.; project administration, A.P.; funding acquisition, S.D. All authors have read and agreed to the published version of the manuscript.

**Funding:** This research received no external funding.

**Institutional Review Board Statement:** Not applicable.

**Data Availability Statement:** No new data were created for this review.

**Conflicts of Interest:** The authors declare no conflicts of interest.

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
