# Peer review of "Advancing Care in Severe Asthma: The Art of Switching Biologics"

_arm, doi:10.3390/arm92020014_

Round 1

Reviewer 1 Report

Comments and Suggestions for Authors

This is a review on switching biologic therapy in patients with severe asthma. The authors make an adequate review of the published studies from which they extract several ideas to recommend to the medical community but, they do not present a direct relationship between the parameters studied and the decision to switch biologics whic is one of the highlights.
It is an adequate review.

Author Response

This is a review on switching biologic therapy in patients with severe asthma. The authors make an adequate review of the published studies from which they extract several ideas to recommend to the medical community but, they do not present a direct relationship between the parameters studied and the decision to switch biologics whic is one of the highlights.
It is an adequate review.

R- Many thanks for your very constructive comment. We have now deeply modified and improved Section 3 (Rationale for switching biologics) by including a direct relationship between the parameters studied and the decision to switch biologics.

Reviewer 2 Report

Comments and Suggestions for Authors

The review, "Advancing care in severe asthma: the art of switching biologics" outlines the rationale for changing biologic therapies in asthma and reviews several studies that have examined patient cohorts after a change in biologic agent.  It provides a useful reference for physicians caring for asthmatics.  This is an important issue for both pulmonologists and general physicians and so this manuscript deals with a clinically important subject.  However, the manuscript has several limitations, which are outlined below.

1. The manuscript would be greatly strengthened by adding a table outlining the differences between the current biologics used in asthma.  As either part of this table or as a separate table, the authors should provide more detail on the most common and/or troubling side effects of the currently available biologics.

2. The review of prior studies needs to include greater detail about the prior studies.  For example, the phrase "poor control" is used as the rationale for switching the meds, but no information is given regarding how the researchers defined "poor control".  Also, there should be greater consistency in the report of the effects of switching.  While the discussion of prior studies does sometimes outline specific findings from the study, the metrics used to evaluate the effects of the switch should be consistently outlined for each study.  If the study under discussion did not report a specific measure used to evaluate the effects of switching to an alternative biologic, then that should be explicitly stated. 

3. Many of the studies outlined in this manuscript that exam switches between biologic therapies report on the effects seen after changing omalizumab to a different biologic.  This current manuscript would be strengthened if the criteria used to start omalizumab were provided for these studies.  For example, one study might have used elevated IgE levels, but another might have used the number of exacerbations in the past year.  This will make it easier for the interested reader to compare these studies.  

4. None of the reported studies exam the effects of switching to tezepelumab-ekko.  Since this is a fairly new biologic there may not yet be studies examining switching and tezepelumab, but the authors should carefully search to confirm that no reports have examined this switch to date before leaving out this new medication.  If no reports can be identified, this should be explicitly stated by the authors.  This is particularly important since the TSLP target is specifically mentioned by the authors in the Abstract.  

5 The "Molecular and immunologic considerations" section is too general.  The authors need to add specific examples to support their recommendations, such as including reported receptor polymorphisms that are associated with differential responsiveness to biologic therapies.  Additionally, the authors should include details regarding the occurrence of specific anti-drug antibodies in patients receiving biologic therapy.

5. Line 233 outlines that patients with 150 eosinophils/microliter before starting dupilumab and 300 eosinophils/microliter before any biologic therapy had better responses.  This is unclear.  Were the examined responses the same for patients starting any biologic and dupilumab?  Do the authors mean to say that all patients with at least 300 eosinophils/microliter before any biologic had very good response to their therapy but patients with the lower eosinophil count of 150 eosinophils/microliter who were started on dupilumab also had very good responses?  

6. The unmasking/development of EGPA after dupilumab described by Eger and colleagues is an important problem and should not be reviewed as part of the overall analysis of studies investigating changes in biologic therapies.  Instead, the problems and significant adverse effects encountered during the medication changes should be discussed in a separate section.

Comments on the Quality of English Language

The manuscript should be revised to be more concise.  Currently, it is both verbose and redundant.  For example, the sections on "Rationale for switching biologics" and "Patient-centered approach" outline similar considerations regarding therapy choice.  

Author Response

The review, "Advancing care in severe asthma: the art of switching biologics" outlines the rationale for changing biologic therapies in asthma and reviews several studies that have examined patient cohorts after a change in biologic agent.  It provides a useful reference for physicians caring for asthmatics.  This is an important issue for both pulmonologists and general physicians and so this manuscript deals with a clinically important subject.  However, the manuscript has several limitations, which are outlined below.

1.The manuscript would be greatly strengthened by adding a table outlining the differences between the current biologics used in asthma.  As either part of this table or as a separate table, the authors should provide more detail on the most common and/or troubling side effects of the currently available biologics.

R- Many thanks for your suggestions. We have now added a new table accordingly (see new Table 1).

  1. The review of prior studies needs to include greater detail about the prior studies.  For example, the phrase "poor control" is used as the rationale for switching the meds, but no information is given regarding how the researchers defined "poor control".  Also, there should be greater consistency in the report of the effects of switching.  While the discussion of prior studies does sometimes outline specific findings from the study, the metrics used to evaluate the effects of the switch should be consistently outlined for each study.  If the study under discussion did not report a specific measure used to evaluate the effects of switching to an alternative biologic, then that should be explicitly stated. 

R- Thanks for this very interesting comment. Following your suggestions, as well as those from other reviewers, we have now deeply improved section 4 (existing literature..) and a new related table.

  1. Many of the studies outlined in this manuscript that exam switches between biologic therapies report on the effects seen after changing omalizumab to a different biologic.  This current manuscript would be strengthened if the criteria used to start omalizumab were provided for these studies.  For example, one study might have used elevated IgE levels, but another might have used the number of exacerbations in the past year.  This will make it easier for the interested reader to compare these studies.  

R- R- Thanks for this very interesting comment. Again, following your suggestions, as well as those from other reviewers, we have now deeply improved section 4 and a new related table

  1. None of the reported studies exam the effects of switching to tezepelumab-ekko.  Since this is a fairly new biologic there may not yet be studies examining switching and tezepelumab, but the authors should carefully search to confirm that no reports have examined this switch to date before leaving out this new medication.  If no reports can be identified, this should be explicitly stated by the authors.  This is particularly important since the TSLP target is specifically mentioned by the authors in the Abstract.  

R- Thanks for your very valuable comment. Due to its novelty, currently there are no data available about switching to Tezepelumab. We have now added in section 7 (future direction) a detailed discussion about how this new biologic may change our approach.

5 The "Molecular and immunologic considerations" section is too general.  The authors need to add specific examples to support their recommendations, such as including reported receptor polymorphisms that are associated with differential responsiveness to biologic therapies.  Additionally, the authors should include details regarding the occurrence of specific anti-drug antibodies in patients receiving biologic therapy.

R- Once again, thanks for your meticulous work. Following your advices, we have now improved section 5 (Molecular and Immunological Considerations) accordingly.

  1. Line 233 outlines that patients with 150 eosinophils/microliter before starting dupilumab and 300 eosinophils/microliter before any biologic therapy had better responses.  This is unclear.  Were the examined responses the same for patients starting any biologic and dupilumab?  Do the authors mean to say that all patients with at least 300 eosinophils/microliter before any biologic had very good response to their therapy but patients with the lower eosinophil count of 150 eosinophils/microliter who were started on dupilumab also had very good responses?  

R- Thanks for this very good comment. We also have now addressed the above in the new version of section 4.

  1. The unmasking/development of EGPA after dupilumab described by Eger and colleagues is an important problem and should not be reviewed as part of the overall analysis of studies investigating changes in biologic therapies.  Instead, the problems and significant adverse effects encountered during the medication changes should be discussed in a separate section.

R- We agree with you. Therefore, following your advice, we have now added a new section addressing the above topic.

Reviewer 3 Report

Comments and Suggestions for Authors

This study has a very promising objective and contains some interesting data, however it requires some improvements to facilitate understanding of the information provided in the article.

1. The first 2 pages are based on 5 articles, but mainly on articles 2 and 3. It would be advisable to write these pages with contributions based on direct sources.

2. The objective of Figure 2 needs more clarity, its relevance is not evident, because it could represent its conclusions, but it is at the beginning of the work.

3. It is also recommended a table on the studies analyzed that would allow you to visualize your findings.

Author Response

This study has a very promising objective and contains some interesting data, however it requires some improvements to facilitate understanding of the information provided in the article.

  1. The first 2 pages are based on 5 articles, but mainly on articles 2 and 3. It would be advisable to write these pages with contributions based on direct sources.

R- Many thanks for your comment. Following your suggestions, we have modified sections 1 and 3, being now less heavily reliant on only articles 2 and 3.

  1. The objective of Figure 2 needs more clarity, its relevance is not evident, because it could represent its conclusions, but it is at the beginning of the work.

R- Since there’s no Figure 2 in the manuscript, we believe that reviewer refers to Figure 1. Thank you for your valuable feedback regarding the objective of Figure 1 and its placement within the manuscript. We acknowledge the concern that the figure's current position at the beginning of the work may not clearly convey its intended message and relevance to the overall narrative. We have now replaced it before "Future Directions" so that it can serve as a visual summary of the considerations involved in switching biologics, tying back to the discussions in the manuscript and leading into future considerations in biologic therapy for severe asthma. Moreover, we have revised caption to provide a clear description of what the figure represents and how it relates to the manuscript's content.

  1. It is also recommended a table on the studies analyzed that would allow you to visualize your findings.

R- Thanks for your very valuable advice. We have now added a new table accordingly (see Table 2).

Round 2

Reviewer 2 Report

Comments and Suggestions for Authors

The revisions made by the authors have addressed the issues raised by my prior review.  The authors are to be commended for their thorough approach to the revisions.

Reviewer 3 Report

Comments and Suggestions for Authors

None